

# Base cations in the soil bank. Non-exchangeable pools may sustain centuries of net loss to forestry and leaching.

Nicholas P. Rosenstock[1], Johan Stendahl[2], Gregory van der Heijden[3], Lars Lundin[4], Eric McGivney[5], Kevin Bishop[4], Stefan Löfgren[4].

[1]Center for Environmental and Climate Research, Lund University, Lund, 232 62, Sweden
[2] Department of Soil and Environment, Swedish University of Agricultural Sciences, Box 7014, 750 07 Uppsala, Sweden
[3]INRA UR 1138, Bioge´ochimie des Ecosyste`mes Forestiers, FR-54280 Champenoux, France
[4]Department of Aquatic Sciences and Assessment, Swedish University of Agricultural Sciences, P.O. Box 7050, 750 07 Uppsala, Sweden
[5]Department of Environmental Science and Analytical Chemistry (ACES), Stockholm University, 106 91 Stockholm, Sweden

*Correspondence to*: Nicholas P. Rosenstock (rosenstockn@gmail.com)

**Abstract.** Accurately quantifying soil base cation pools is essential to interpreting the sustainability of forest harvests from element mass-balance studies. The soil exchangeable pool is classically viewed as the bank of "available" base cations in the soil, withdrawn upon by plant uptake and leaching and refilled by litter decomposition, atmospheric deposition and mineral weathering. The operational definition of this soil bank as the exchangeable (salt-extractable) pools ignores the potential role of "other" soil nutrient pools, including microbial biomass, clay interlayer absorbed elements, and calcium oxalate. These pools can be large relative to "exchangeable" pools. Thus neglecting these other pools in studies examining the sustainability of biomass extractions, or need for nutrient return, limits our ability to gauge the threat or risk of unsustainable biomass removals. We examine a set of chemical extraction data from a mature Norway spruce forest in central Sweden, and compare this dataset to ecosystem flux data gathered from the site in previous research. The 0.2 M HCl extraction released large pools of Ca, K, Mg, and Na, considerably larger than the exchangeable pools. Where net losses of base cations are predicted from biomass harvest, exchangeable pools may not be sufficient to support more than a single 65-year forest rotation, but acid-extractable pools are sufficient to support many rotations of net-ecosystem losses. We examine elemental ratios, soil clay and carbon contents, and trends with depth of these pools to identify the likely origin of the HCl-extractable pool. No single candidate compound class emerges as very strongly supported by the data to be likely to constitute the majority of the HCl-extractable fraction; a combination of microbial biomass, fine grain, potentially shielded, easily weatherable minerals, and non-structural clay interlayer bound potassium may explain the size and distribution of the acid extractable base cation pool. Sequential extraction techniques and isotope exchange measurements should be further developed and, if possible, complemented with spectroscopic techniques to illuminate the identity of and flux rates through these important, and commonly overlooked, nutrient pools.



# 1 Introduction

## 1.1 Importance of soil base cation pools for sustainable land use.

In an attempt to decrease net $CO_2$ emissions and to meet the increasing demand for woody biomass products and energy, some forest policies encourage the intensification of biomass extraction from forest ecosystems (Royo et al., 2012; Lauri et

al., 2014; Kazagic et al., 2016). These practices may severely impact soil fertility, especially in forest ecosystems where nutrient availability may be low (Feller, 2005; Kreutzweiser et al., 2008; Thiffault et al., 2011; Achat et al., 2015). For example, in the USA it has been projected that increased biomass harvest for biofuels will markedly increase the area of managed forest that will experience net nutrient losses due to biomass export: from 17 to 50% of the total forest area for Ca, 20 to 57% for K and 16 to 45% for Mg (de Oliveira Garcia et al., 2018). Similar concerns have been raised in other

countries, including Sweden (Akselsson et al., 2007a), Belgium (Vangansbeke et al., 2015), Finland (Aherne et al., 2012) and Germany (Knust et al., 2016). Quantifying sustainable levels of biomass extraction, such that soils are not progressively impoverished and productivity can be maintained without nutrient inputs, is therefore a major concern for forest managers and policy makers (Lucas et al., 2014; Achat et al., 2015). The pressures on plant-available base cations result not only from increased harvest intensity, but also from continued atmospheric nitrogen and sulfur deposition (Iwald

et al., 2013; McGivney et al., 2019), and elevated atmospheric concentrations of $CO_2$ (Duval et al., 2012; Terrer et al., 2016).

The nutrient mass balance (*i.e.* input-output budget) approach is commonly used to estimate the net gain or loss of nutrient elements in the soil or ecosystems under different management and climate scenarios (Akselsson et al., 2007a; Knust et al., 2016; de Oliveira Garcia et al., 2018). This mass-balance approach typically accounts for inputs via

atmospheric deposition and the weathering of primary minerals as well as secondary clay minerals, and exports occur via leaching and biomass removals (Nilsson *et al.*, 1982; Van Breemen *et al.*, 1984). Under this definition, the mass balance equation estimates the change in nutrient content in a "source and sink" reservoir of the soil. For base cations, this "source and sink" reservoir is assumed to be the pool of Mg, Ca, Na and K stored in the soil as exchangeable cations adsorbed on the cation exchange complex. This quantity is conventionally measured by ion-exchange soil extractions using

concentrated salts (salt-extractable exchangeable pools; Akselsson et al., 2007a; Vangansbeke et al., 2015; Knust et al., 2016).

Accurate estimates of soil nutrient stocks are important for policies on sustainable levels of biomass extraction levels or critical loads, i.e. the maximum level of atmospheric pollutant deposition that will not damage sensitive aspects of the ecosystem, as well as for determining the maximum level of forest biomass harvest, the "critical biomass harvest" that

will not result in unacceptable soil and stream acidification (Akselsson & Belyazid, 2018). The net balance of nutrients between influx and efflux in an ecosystem or soil can only be meaningfully interpreted in the context of the magnitude of soil nutrient reserves. If, for example, whole-tree harvesting results in a net decrease of soil Ca, it is equally important to know if this loss is likely to result in significant reductions in the availability of Ca to plants in 1, 10 or 100 forest rotations as it is to know whether that balance is positive or negative.

## 1.2 Use of salt-extractable pools in ecosystem mass-balance studies

Base cation nutrient reserves in soil are commonly estimated by measuring salt-extractable quantities, commonly with barium and ammonium salts as extractants, to represent what is available to plants (Skinner et al., 2001; McLaughlin and Philips, 2006; Brandtberg and Olsson, 2012; Zetterberg et al., 2016). However, on timescales of years to decades, salt-extractable base cation reserves in the soil do not appear to accurately predict what is available for leaching or biological

uptake (Mengel and Rahmatullah, 1994; Bailey et al., 2003; Hamburg et al., 2003; Lucash et al., 2012). This may be due to the presence of other significant pools of base cations in the soil, which are not salt-extractable, but which contribute to the



plant-available pools of base cations. Calcium in known to form insoluble complexes with organic compounds such as oxalate anion, which are not salt-extractable (Dauer et al. 2014), but these complexes may be available to plants and microbes over relatively short timescales. The microbial biomass of the soil is quite large, has considerably higher nutrient concentrations than bulk organic matter, and turns over very rapidly, and may thus represent a significant pool of available

base cations that is missed in salt extractions (Yamashita et al., 2014, Lorenz et al. 2010, van der Heijden et al. 2014). Aluminium and iron oxides and hydroxides may also directly or indirectly (via organic matter or phosphate bridges) adsorb base cations (Kinniburgh et al., 1976; Grove et al., 1981). For example, in a highly weathered tropical soil, Hall and Huang (2017) showed that a significant amount of Mg, Ca and K was sequestered in Fe (hydr)oxide secondary mineral phases and that the microbial dissolution of these phases increased the concentrations of weak-acid extractable cations.

Significant stocks of soil potassium may be stored in the interlayers of clay minerals (Sparks, 1987). This pool of interlayer K is not accounted for by salt-extractable exchangeable cations but may be quite significant for plant nutrition: Falk Øgaard and Krogstad (2005) showed that interlayer K accounted from 26% to 43% of K uptake in grassland ecosystems. Not accounting for these additional pools of base cations in the soil may explain why mass balance models often fail to reproduce the empirically measured change in salt-extractable exchangeable pools (Löfgren et al 2017; van der Heijden et

al. 2014).

Measurements of salt-extractable base cations may be complemented by soil extractions using strong acids. Aqua Regia, HF, or lithium metaborate fusion extract the total or near total reserves of elements, which may then be used to i) estimate the relative distribution of minerals in the soil (Posch and Kurz, 2007), and ii) estimate the mineral weathering flux based on assumptions of weathering kinetics derived from laboratory dissolution experiments (Warfvinge and

Sverdrup, 1992). Moderately concentrated (defined here as 0.1 M -1 M) strong acid extractions (primarily HCl, but also $HNO_3$) have been used to extract nutrient elements from soil organic matter (e.g. Ca from Ca-oxalate; Dauer et al. 2014), secondary minerals (e.g. K from clay interlayers; Simonsson et al., 2016) and relatively labile or easily weathered primary minerals (e.g. Ca from apatite, Mg from biotite; Blum et al., 2002; Lucash et al., 2012).  However, these extractions are non-specific, and will dissolve a range of secondary minerals, likely release microbial nutrient pools, and partially dissolve

non-target granite-derived tectosilicate minerals.

### 1.3 Study objective

In order to examine the potential importance of non-exchangeable sources of base cations in relation to net losses of base cations from forest harvesting we utilized published data from a particularly well-studied forest site at Kindla, Sweden to form a mass balance budget of base cations under different harvest scenarios, and compared annual losses, where

predicted, to base cation pools in the soil as defined by different extractants. The potential for stem-only or whole-tree harvesting to result in net losses of base cations was examined, then these estimated losses were compared to extraction-defined (distilled water, $BaCl_2$ 0.1 M, HCl 0.2 M, HCl 0.5 M, and Aqua Regia) base cation pools to estimate the number of harvest rotations of net nutrient losses these pools could potentially buffer against.  The aim of this paper is to discuss i) what additional pools of base cations in the soil may actively contribute to the "source and sink" pool, ii) how important in

terms of size these additional pools can be and iii) how these additional pools may contribute to replenishing plant-available pools over time and under different management scenarios.

## 2 Methods

### 2.1 Study area

Kindla is a catchment within the Swedish national integrated monitoring program (Löfgren et al 2011) and a core site in

LTER Europe (https://deims.org/search/all/Kindla). The monitoring was initiated in 1995 and is related to the Convention

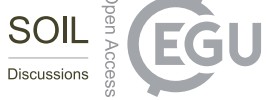



"Long-Range Transboundary Air Pollution – LRTAP 1979" (UN/ECE) and the International Cooperative Program on Integrated Monitoring of Air Pollution Effects on Ecosystems (ICP-IM, https://www.unece.org/env/lrtap/workinggroups/wge/im.html). It focuses specifically on the monitoring of physical, chemical as well as biological processes in time and space to increase the knowledge on causes of ecosystem changes

(Starr, 2011). Data from the Kindla catchment are used for biogeochemical process evaluations (see e.g. articles in Starr, 2011), status and trend assessments (e.g. Vuoremaa et al. 2018) as well as model exercises (e.g. Zetterberg et al. 2014, McGivney et al. 2019). The Kindla site is also designated as one Critical Zone Observatory and in this article, we evaluate data from Kindla soils collected and analysed within the EU Fp7 project Soil Transformations in European Catchments (SoilTrEC, Banwart et al. 2011). The study was initiated due to lack of forest harvest induced depletion of the exchange Ca

and Mg pools even though mass-balances calculations indicated such declines should occur (Olsson & Lundin, 1990).

The forest examined at Kindla is a naturally regenerated, uneven-aged Norway Spruce (*Picea abies*, L.) forest, which has been minimally affected by forestry activities during the last century. Granitic bedrock is overlain by till soils of the same geological origin. Soils from three proximate forest plots were sampled from the Kindla catchment (59∘05′ N 12∘01′ E; Löfgren *et al.*, 2011). The three plots cover an elevation gradient from 300-320 m.a.s.l. over a distance of ~150

meters. The upper and intermediate sites are in groundwater recharge areas on Podzols and Regosols, respectively. The lowermost site is also located on a Regosol in a groundwater discharge area with considerably more organic matter in the upper mineral horizons (See sup. methods and Table 1 for more detailed description).

### 2.2 Base cation fluxes and forest mass-balance

For each plot and harvest scenario the mass balance of each base cation, Ca, K, Mg, and Na, was calculated as:

$$BC_{bal} = BC_{dep} + BC_{weath} - BC_{leach} - BC_{uptake}$$

where $BC_{dep}$ represented inputs via atmospheric deposition (wet and dry), $BC_{weath}$ represented inputs to the soil via mineral weathering, $BC_{leach}$ represented losses via leaching below the B horizon, and $BC_{uptake}$ represented uptake of base cations into vegetation and subsequent removal at harvest, all terms were calculated per $m^2$ per year. Total deposition (wet+dry) of base cations and mineral weathering rates were taken from McGivney et al. (2019). Base cation uptake into tree biomass and removal via either stem-only (stems + bark) or whole-tree (roots+stump+stem+bark+branches-logging losses) harvest

were taken from Zetterberg *et al.* (2014), McGivney *et al.* (2019) and Nilsson et al. (2018). Leaching estimates were based on soil water sampled with suction lysimeters (P80 ceramic cups) in the B horizon and compared to catchment runoff (Löfgren et al. 2011). Mineral weathering inputs were taken from modelled values from McGivney *et al.* (2019). Deposition, weathering, and biomass uptake fluxes were available for the entire catchment including the three soil sampling sites located along a hydrological gradient (recharge, intermediate and discharge areas), while leaching losses

were calculated for each of these three hydrological locations (For a more detailed description on how each flux was calculated see sup. methods).

### 2.3 Soil base cation pools

As part of the EU FP7 funded SoilTrEC project (Banwart et al. 2011) a series of non-sequential chemical extraction were performed on 9 composite mineral soil samples, representing the E, B, and C horizons at the three soil sampling sites.

Extractions were performed with distilled water, 0.1 M BaCl$_2$, 0.01 M ammonium lactate, 0.2 M HCl, 0.5 M HCl, and Aqua Regia (3HCl:HNO$_3$). These extractions were filtered and measured for elemental contents of base cations Ca, Mg, K, Na as well as P, Fe and Mn. Soil pH, carbon and nitrogen concentrations as well as soil texture and moisture content were also measured for each sample (see Table S1 and sup. methods for more detailed description). The O horizon was not included in the SoilTrEc project, but the organic horizon of the same hillslope was in 2006 sampled within the EU Life+

project BioSoil (De Vos and Cools, 2011) and extracted with 0.1 M BaCl$_2$, and Aqua Regia. The SoilTrEC and BioSoil



soil sampling pits are not identical, but forest age, structure and management history, hydrologic gradient, soil types, humus layer thickness and bulk density are identical. Hence, we have used the BioSoil extraction data for estimating these two base cations pools in the organic horizon.

For the soils, elements, and harvest scenarios that have a net elemental loss over time, we examined the potential for different soil base cation pools to sustain those losses. We assumed that ecosystem fluxes (deposition, weathering, leaching, biomass uptake) would remain constant, and calculated the number of 65-year harvest rotations that each extractant-defined base cation pool could offset net ecosystem losses for each harvest scenario–element-soil combination under which a net ecosystem loss was predicted.

## 3 Results and Discussion

### 3.1 Soil base cation pools

In the mineral soil, concentrations of Ca and Mg in $H_2O$ extractions were generally far less ( < 1/10) then those in the $BaCl_2$ extractions, while $H_2O$ extractions of K were only slightly less, and $H_2O$ extractions of Na were slightly greater than $BaCl_2$ extractions (Fig. 1). Ammonium lactate extractions of all base cations were similar to $BaCl_2$ extractions; $BaCl_2$ extracted nearly the same or slightly more of the bivalent Ca and Mg, while ammonium lactate tended to extract more monovalent K and Na (100% - 150%). $BaCl_2$ exchangeable pools of Ca, K, Mg, and Na were particularly high in the E horizon of the downslope Regosol in the groundwater discharge area with significantly higher SOC content. Within each soil, $BaCl_2$ exchangeable cations tended to decrease with increasing depth. The extracted amounts of Ca, K, and Mg in the 0.2M HCl extraction were considerably (5-100 fold) greater than the amounts obtained from $BaCl_2$ extraction. As compared to the 0.2 M HCl extraction, 0.5 M HCl extraction yielded approximately 2-fold greater release of Ca, but only about 1.3-fold greater release of K, Mg, and Na. HCl extraction yields of Ca and Mg increased significantly with depth, while HCl extractions of K did not display a depth trend, and on the upslope and midslope soils E horizon yields of K were higher than from the B horizons. HCl extraction yields of Na exhibited a slight decreasing trend with soil depth. Compared with the 0.5 M HCl extraction yields, Aqua Regia extracts of Mg and K were approximately 2-fold greater and Aqua Regia extracts of Ca approximately 6-fold greater, while Aqua Regia extraction yields of Na were slightly lower (3%) than from 0.5 M HCl. Aqua Regia extractions of all base cations increased with depth; this depth trend was strongest for Ca and Mg, weaker for K, and almost absent for Na.

Humus layer 0.1 M $BaCl_2$ extractable pools of Ca, K, Mg, and Na were 88, 100, 86, and 99%, respectively, as large as Aqua Regia extracted pools, indicating that nearly all of the humus-associated base cation pools are exchangeable. Humus layer 0.1 M $BaCl_2$ extractable pools of Ca, K, Mg, and Na were 2, 5, 4, and 0.6 times as large as the total mineral soil 0.1 M $BaCl_2$ extractable pools, indicating a predominant role of the humus layer as a reservoir of salt-extractable base cations. Considering only the mineral soil sampled in this study, the $BaCl_2$ extractable pools of Ca. K, Mg, and Na were 12, 5, 1, and 6%, respectively, of the size of the 0.2 M HCl pools. If we include the humus layer base cation pools then the $BaCl_2$ extractable pools of Ca. K, Mg, and Na increase to 20, 24, 3, and 9%, respectively, of the size of the 0.2 M HCl extractable pools. Hence, 3-24% of the base cations are found in the salt extractable $BaCl_2$ pool, while the remaining 76-97% is found in the 0.2 M HCl pool.

### 3.2 Ecosystem base cation mass-balance budgets

Inputs of Ca, K, and Na via weathering were 1.8-, 1.5-, and 1.3-fold greater, respectively, than inputs via atmospheric deposition, while deposition inputs of Mg were 2-fold greater than weathering inputs. Leaching losses of Ca and Mg were highest in the downslope Regosol from the groundwater discharge area, and leaching losses were less than the combined inputs of weathering and deposition for Ca, K, and Na, but not for Mg (Table 2). Whole-tree harvesting resulted in





considerably greater elemental losses than stem-only harvest (2.3 – 2.6 fold greater). Whole-tree harvesting resulted in net loss of Ca, K, and Mg on all three plots, while stem-only harvesting resulted in net losses of only Mg on all three plots and net losses of Ca on the downslope Regosol (Table 2).

**3.3 Base cation depletion over time**

Based solely on the mineral soils, the exchangeable (BaCl$_2$-extractable) pools of Mg were not sufficient to sustain a single 65-year rotation of stem-only or whole-tree harvest (Mg$_{exch}$ was sufficient to support 1.3 rotations on the downslope soil; Table 3). Exchangeable Ca pools were sufficient to sustain 0.2, 0.8, and 1.5 rotations, and K pools were sufficient to sustain 0.6, 2.0, and 5.3 rotations, of whole-tree removal on the upslope, midslope, and downslope soils, respectively (Table 3). If the humus layer BaCl$_2$-extractable base cation pools are also considered, the number of rotations of stem-only

harvest which could be sustained by the BaCl$_2$-extractable Mg pool increases to 2.5 – 3.1 depending on hillslope location. For the whole-tree harvest scenario, this range increases to 1.1 – 2 rotations for the BaCl$_2$-extractable Ca and Mg pools, while the number of rotations which could be sustained by the BaCl$_2$-extractable K pool increases along the hillslope gradient to 2.3, 15.5, and 31.4 65-year rotations, with the highest value downslope. Were it available for plant uptake, the 0.2 M HCl extractable base cation pools would be able to sustain considerably more rotations of net ecosystem losses

predicted under whole-tree harvesting than the BaCl$_2$-extractable pools. The mineral soil 0.2 M HCl-extractable pools of Mg were sufficient to sustain at least 37, and the 0.2 M HCl-extractable pools of Ca at least 7, rotations of whole-tree removal. The 0.2 M HCl-extractable pools of K pools were sufficient to sustain 8, 65, and 85 rotations of whole-tree removal on the upslope, midslope, and downslope soils, respectively (Table 3).  The number of rotations that could be sustained by the 0.5 M HCl extractable base cation pools was 15 - 90% greater than by the 0.2 M HCl-extractable pools,

while the total amounts of base cations, determined by the Aqua Regia extraction, could sustain at least 57 whole-tree harvesting rotations (Table 3).

**3.4 Potential sources of acid-extractable base cations in the mineral soil**

Due to lack of data from the organic horizon, this chapter is primarily focused on the results from the mineral soils. Given the large size and potential to buffer against leaching and many forest harvest rotations of net base cation loss,

understanding the chemical nature and availability of the HCl-extractable pool is important to forming management recommendations. Extractions with strong acids at moderate (0.05 M – 1 M) concentrations, as was performed in this study, have been used to selectively extract calcium oxalate (Dauer and Perakis, 2014; Cromack et al, 1979), apatite (Blum et al., 2002; Nezat et al., 2007), nonexchangeable clay interlayer K (Simonsson et al., 2016; Li et al., 2015), and Fe- and Mn-oxides (Krasnodębska-Ostręga et al., 2001) from soils.

Despite the prevalence of their use, the use of HCl and HNO$_3$ at room temperature or above, from 0.01 M to 2 M strength, as extractants, are not specific to particular chemical compounds or soil minerals, as all primary and likely all secondary minerals and organocomplexes of Ca, K, Mg, and Na are susceptible to proton-promoted dissolution (Skyllberg *et al.*, 2001), albeit to varying degrees. Because these acid extractions solubilize a range of compounds from mineral soils, elucidating the predominant physicochemical origin of the base cations in the very large HCl-extractable portion is

challenging. It would aid our understanding of the potential availability and flux rates of base cations to know, for example, whether the HCl-extractable pool is comprised of incompletely weathered minerals, either as weathering rinds remaining after non-stoichiometric dissolution or secondary mineral coatings that form during weathering, or if it is comprised of secondary compounds that formed after base cations have entered the soil solution via weathering reactions (and may thus also be fed by plant derived base cations). By examining the base cation ratios and amounts in different





extractions we may be able to draw some conclusions about the origins and from which pools the sizable HCl-extractions of base cations derive from.

### 3.4.1 Microbial biomass

Microbial biomass has been suggested to be an important reservoir of N and P, and potentially of base cations. We estimated the potential size of the mineral soil microbial biomass base cation pools as a function of soil carbon and microbial biomass carbon to nutrient ratios (for more detail see supp. methods). Microbial biomass was estimated to constitute a small but potentially significant fraction of Ca, Mg and K pools, particularly K. Excluding the humus layer, which is dominated (88-100%) by the $BaCl_2$ extractable base cation pools, the potential microbial biomass pool of Ca was 5-36% as large as the $BaCl_2$-extractable pool, and the potential microbial biomass pool of K was considerably larger than the $BaCl_2$ extractable K pool (by a factor of 1.2 - 4.2) for the 2 lower Regosol soils, and 26-90% of the $BaCl_2$ extractable K pool at the upper Podzol site. If we assume that microbial biomass contents were extracted in the 0.2 M HCl and stronger acid extractions, microbial biomass constituted on average 2% and 16% of 0.2 M HCl extractable Ca and K pools, respectively. In the downslope Regosol, which lies in a groundwater discharge area and had much higher SOC contents, microbial biomass may have contributed 4% and 26% of 0.2 M HCl extractable Ca and K pools, respectively (Fig. 2; Table 4).

### 3.4.2 Differences between acid-extracted base cation pools

If we assume that both the 0.2 M and 0.5 M HCl extractions extract all, or nearly all, of the $BaCl_2$-extractable and microbial biomass fractions in mineral soil, and assume that the Aqua Regia extraction will also extract all the base cations that the 0.5 M HCl extraction would, then we can examine the differences between these extractions as being more diagnostic of the mode of chemical attack of the more aggressive (in terms of greater extraction yield) extractant. We will use the term "only" to designate base cation pool estimates that are derived from subtracting extracted amounts from one extractant pool from another. Thus, the 0.2 M HCl-only pool is what remains of the 0.2 M HCl extractable base cation pools after both the estimated microbial biomass and $BaCl_2$-extractable base cation pools were subtracted; 0.5 M HCl-only is the difference between the 0.5 M HCl- and 0.2 M HCl-extractable pools; Aqua Regia-only is the difference between the Aqua Regia and 0.5 M HCl extractable pools. We can further examine the molar ratios of base cations in these pools and compare both BC pool sizes and elemental stoichiometry as well as examine potential relationships between these and clay and organic matter contents (Table 1).

The pool of Aqua Regia-only Ca is far larger (4-5-fold) than either HCl-only extractable Ca pool (Table 4; Fig. 2). This stands in contrast to K, Mg, and Na, for which the 0.2 M HCl-only pools are nearly as large, or larger than the Aqua Regia-only fraction. Calcium also stands out in that the 0.5 M HCl-only and 0.2 M HCl-only digestible fractions were similar in size while, in contrast, for K, Mg, and Na the stronger acid yielded relatively less additional cations (less than half as much as the 0.2 M HCl-only). Calcium is the dominant cation (comprising 50% or more of total base cations, on a molar basis) in both the exchangeable and Aqua Regia base cation pools, but in the HCl extractable pools it comprised a smaller portion (Table 4). In contrast we see for Mg that the two HCl-only pools combined are larger than Aqua Regia only pools of Mg, and we also see that the mole fraction of base cations is heavily enriched for Mg in the HCl extractable fraction. HCl-extractable Mg also displays a strong depth trend in each of the 3 soils, being 5-70 times as abundant in the C horizon HCl-only extracts as compared to E-horizon extracts. While the amounts of both Mg and Ca increase with depth in the HCl-extractions, HCl-extractable K does not display the same behaviour; in the upslope and midslope soils its abundance is relatively stable with depth, and the relative abundance of K ($[K]/[BC_{tot}]$) is notably higher in the uppermost E horizon.



### 3.4.3 Base cations bound to soil organic matter

Base cations may be bound to organic compounds in the soil and organically-bound base cations may represent a significant pool of exchangeable cations (Duchesne and Houle, 2008; Richardson et al., 2017). The humus layer 0.1 M $BaCl_2$ extractable pools of Ca, K, Mg, and Na were 88, 100, 86, and 99%, respectively, as large as the humus-layer Aqua

Regia extracted pools. Focussing on the divalent ions calcium and magnesium, these results as well as earlier studies (Bailey et al., 2003; Dauer and Perakis, 2013) show that a significant fraction of the organically bound cations may not be extractable with concentrated salts. Calcium e.g. may form strong complexes with organic acids (Tipping 2002), such as calcium oxalate. Calcium oxalate has been suggested to accumulate in soils over time, potentially in large amounts (Cromack et al., 1979; Dauer and Perakis, 2013).  If organically-bound calcium were a significant portion of the HCl-

extractable pools, we might expect to see a decrease with depth as Ca-oxalate is most associated with exudation and incomplete decomposition of organic matter near the soil surface, in addition we might expect to see a relative increase in the $Ca:BC_{tot}$ elemental ratios, especially near the soil surface. Instead we observed that $Ca:BC_{tot}$ are relatively depleted in the HCl-extractable fractions compared to both the exchangeable and aqua-regia extractable fractions; in addition, we see no change or a slight increase in the $Ca:BC_{tot}$  ratios with depth in the HCl-extractable pools. Moreover, we would expect

organically-bound Ca to be entirely dissolved by the 0.2 M HCl extraction, and thus, if it dominated the HCl-only pools, we would expect considerably less Ca in the 0.5 M HCl-only pool, which we do not find. Finally, if a significant pool of the acid-extractable base cations were locked in organic matter, we would expect a correlation between soil carbon and the acid-extractable BC pools, but we do not. We do see a fairly strong correlation between organic carbon and $BaCl_2$-extractable Ca, Mg, and Na pools (%C vs Ca, $r^2 = 0.56$; vs K, $r^2 = 0.00$; vs Mg, $r^2 = 0.61$;  vs K, $r^2$ 0.71; data not shown),

but not between organic C and acid-extractable BC pools.  Taken together these observations suggest that organically-bound Ca (including Ca-oxalate) is unlikely to be a major component of the HCl-extractable Ca pools. Indeed, Dauer and Perakis (2014) observed that while production rates of Ca-oxalate may be high, Ca-oxalate is also highly susceptible to microbial degradations, such that acid extractable Ca-oxalate pools comprised less than 3% of exchangeable Ca pools.

Soil organic matter (SOM) may also form protective coatings on soil minerals reducing their solubility (Drever

and Stillings, 1997). It is possible that a significant portion of mineral surfaces were shielded with SOM, and the HCl extractions may have removed these protective layers. This could potentially explain why, particularly for K and Mg, we see considerably greater BC yields in the 0.2 M HCl-only fraction than we see in the 0.5 M HCl only fraction, because 0.2 M HCl is sufficient to remove these protective coatings and the resultant burst of mineral surface dissolution is not a function of acid strength, but a function of the removal of these coatings. To examine the potential for OM coatings on

minerals to shield them from weathering, but be readily removed by 0.2 M HCl, chemical treatments utilizing hydrogen peroxide or surfactants (Chao, 1984) could be used to remove or reduce such coatings without the use of a strong acid.

### 3.4.4 Clay interlayer potassium

Non-structural K, strongly bound in clay interlayers has been observed to be present in large quantities in a variety of soils (Moritsuka et al., 2003; Li et al, 2015). Clay interlayer K is typically extracted with dilute or concentrated

$HNO_3$ or HCl, or specific cation-exchange reactants, such as sodium tetraphenylboron (NaTPB). HCl and $HNO_3$ have been shown to release significant portions of this pool, but also to cause significant, though limited, weathering of clays; NaTPB, in contrast has been shown to release far more interlayer K, and not exhibit any observable weathering (Li et al., 2015; Moritsuka et al., 2003). If clay interlayer exchangeable/occluded K were a major storage pool then we might expect the size of the acid-extractable K pool to correlate with clay content, which we do not see. Instead, there is a weak negative

correlation between both the 0.2 M and 0.5 M HCl-extractable K and clay content (as measured by particle size, Table 1). However clay content, as measured by particle size may not accurately reflect the amount of secondary clay minerals, or,



in particular, the amount of minerals which are likely reservoirs of interlayer K (Simard et al., 1989). Indeed, Mengel and Rahmatullah (1994) observed that the clay interlayer K pool was both large and increasingly important for plant uptake in coarser soil mixtures as clay content decreased, due to the relative reduction in exchangeable K, and Simonsson et al. (2016) showed that clay interlayer K may constitute a very large pool of K  ( > 20 X exchangeable K) even when clay

contents were quite low (3-5%). We observed that HCl-extractable K in the E horizon was larger than that of the B horizon and similar in size to the C horizon in 2 of the 3 soils (upslope and midslope). This lack of a clear depth trend, which both Mg and Ca have, in conjunction with the abovementioned high proportion of the total base cation pool which is K in the E horizon, may indicate an accumulation of non-structural clay interlayer K in the upper horizons of the upslope and midslope soils. Use of NaTPB, instead of HCl, or use of repeated sequential extractions of cold 0.01 M HCl (Moritsuka et

al., 2003) may avoid the non-specific dissolution activity of stronger acids and be more diagnostic for extraction of K bound in clay interlayers.

### 3.4.5 Fe- and Mn-oxides

       HCl is commonly used, both as a single extractant and in sequential extractions to dissolve Fe and Mn oxides in soils (Chao and Sanzolone, 1992; Krasnodębska-Ostręga et al., 2001). Soil Fe and Mn oxides can be important reservoirs

for base cations not extractable with typical salt exchange assays (Krasnodębska-Ostręga et al., 2001). Indeed we extracted large amounts of Fe in the 0.2 M HCl- only pool (> 50% total Fe, $0.01 - 0.1$ % total soil dry mass), though relatively little Mn ( ~ 1/100[th] as much as Fe by mass; data not shown). There was no significant correlation between Fe extracted in any of the acid fractions and the contents of any base cations in those fractions. In addition, the HCl-only pools had the lowest $BC_{tot}$:Fe ratios amongst all extraction pools (Table 4). Our data indicate that despite being both readily extractable and

present in the soils in very high amounts, binding to Fe-oxides fractions appears unlikely to account for the majority of the acid-extractable base cation pools.

### 3.4.6 Primary minerals

       While more aggressive digestions (Aqua Regia, HF, $LiBO_2$-fusion) are typically used to estimate elemental contents of primary minerals in soils, the large pools of BC observed in the HCl digestions may have largely arisen from the HCl-

induced dissolution of primary minerals. The Ca, K, and Mg contents of the 0.2 M HCl-only pools correspond to 357, 174, and 6832 years of mineral weathering, given the weathering rates used in this study from McGivney et al. (2019). If this dissolution was congruent and in proportion to the soil mineral contents we might expect similar elemental ratios in the HCl-only extractable pools as we observe in the Aqua Regia pool. We see instead that Ca:$BC_{tot}$ are relatively lower, and K:$BC_{tot}$  and Mg:$BC_{tot}$ are relatively higher in the HCl extractable fraction as compared to the Aqua Regia digestion.

However, incomplete mineral dissolution with HCl and $HNO_3$ at concentrations of 1 M or less acids is known to yield incongruent dissolution, preferentially releasing base cations from primary minerals (Snäll and Liljefors, 2000, Chao, 1984; Moritsuka et al., 2003) and sediments (Agemian and Chau, 1976; Sutherland, 2002) and the distribution of minerals in different soil particle size classes is understood not to be representative of bulk minerology (Simard et al., 1989). The size, depth distribution and relatively higher molar ratio of Mg in the HCl-only pool support the possibility of easily

weatherable Mg-containing minerals as a likely source. Modelled minerology from total soil elemental contents in the Kindla soils indicate the presence of apatite (0.1-0.2%), chlorite (0.2-0.8%) and hornblende (0.3-1.8%) in the E and B horizons (McGivney et al. 2019) and regional databases indicate the presence of these minerals as well as biotite in soils with similar geologic origin (Stendahl et al., 2002). In mineral and soil dissolution studies chlorite, hornblende, and biotite have been shown to yield high quantities of Mg in extractions with 1M HCl (Snäll and Liljefors, 2000), although Mg

extraction efficiency from hornblende were considerably lower and more particle size dependant. For Ca, apatite stands out as a likely mineral source of the HCl-extractable Ca, but were apatite the dominant source of Ca, we might expect a



different relative distribution between the different extractions. As apatite is highly susceptible to proton-promoted dissolution, more so than the aforementioned potential Mg source pools (Nezat et al., 2007), we would expect relatively larger portions of total Ca to have been extracted in the 0.2 M HCl-only vs 0.5 M HCl-only and in the HCl-only vs the Aqua Regia pools than we see. These finding indicate that either significant portions of the Ca are coming from more

recalcitrant minerals than apatite, or that there is significant mineral shielding of apatite, and stronger acids than 0.2 M HCl are needed to scavenge these apatite inclusions. Nezat et al. (2007) observed that the 70% of apatite mineral originally present in the soil was weathered completely in the upper soil layers and the remaining 30% was shielded by inclusion in granitoid minerals and thus inaccessible to acid digestion with 2 M $HNO_3$ (though accessible to acid digestion with HF). Were apatite to be a primary source of Ca in the 0.2 M and 0.5 M HCl-only extractions we might expect Ca:P molar

rations close to 10:6 as is the case for apatite. Using this ratio, assuming congruent dissolution, and assuming all of the P contents in the 0.5 M HCl-only extraction could be attributable to apatite, 20-27% of the C-horizon Ca content in 0.5 M HCl-only extractions could be attributable to apatite.

### 3.4.7 Physicochemical orgin of HCl-extractable base cation pools

Our data indicate that the large pools of 0.2 M HCl-extractable base cation pools may be comprised of a combination of

microbial biomass, the dissolution of fine grain, potentially OM-shielded, easily weatherable minerals (such as biotite, apatite, chlorite, and hornblende), as well as by the presence of significant stores of non-structural clay interlayer bound K. Determining the physicochemical nature of these base cation pools is central to incorporating them into mass-balance and geochemical models widely used today to inform management and policy. If these base cation pools derive from primary mineral weathering then weathering models may already account for these pools or may need to be adjusted to account for

these large and potentially labile pools; if they are a result of organic complexation or secondary minerals, then understanding what controls their formation and availability depends on a better understanding of their chemical nature. Combining sequential extraction approaches with X-ray diffraction methods on small soil samples of different particle size classes may help elucidate the potential importance of primary minerals to the HCl-extractable pools, while other spectroscopic methods (*e.g.* micro, nanoSIMS, STXM, etc.) may be employed to determine the role of organic or

secondary mineral coatings or identify secondary mineral or non-crystalline base cation pools.

### 4 Importance of non-exchangeable or acid-extractable base cation pools to nutrient balance concerns in forests

Among the many studies which have predicted net ecosystem losses of base cations with biomass (stem-only or whole-tree) extractions, exchangeable base cation pools, when measured, are commonly not sufficient to sustain more than a single harvest rotation, assuming productivity, growth, and base cation uptake are not reduced by reductions in base cation

availability (Akselsson et al., 2007a; Akselsson et al., 2007b; Duchesne and Houle, 2008; Knust et al., 2016). We are not aware of any other studies of ecosystem mass-balance of base cations that have attempted to explore the sizes of non-exchangeable nutrient pools in relation to net ecosystem losses over time. Those that have compared HCl- or $HNO_3$-extractable base cation contents in similar soils have commonly found acid-extractable pools to be many fold larger than salt-extractable exchangeable pools (Olofsson et al., 2016; Nezat et al., 2007; Simard et al., 1989; Moritsuka et al., 2002;

Lucash et al., 2012).

The relevance of the size of these large pools of acid-extractable base cations to forest nutrient relations is a function of their flux rates or bioavailability. If the base cations released from 0.2 M HCl-only pools are highly stable over centuries and not available to refill depleted exchange sites from which plant uptake may occur then they are of little relevance to policy decisions on sustainable harvest levels. If, on the other hand, they are in equilibrium with the

exchangeable pools and available for plant uptake in significant amounts on decadal or century time scales then these pools





and their size relative to the exchangeable pool are of immediate relevance to forest management recommendations in light of the immediate need to reduce fossil fuel emissions.

A number of studies have examined the bioavailability and flux into exchangeable pools of acid-extractable non-exchangeable pools of base cations across a variety of soils. Callesen et al. (2004) examined 19 Danish forest soils

comparing exchangeable base cation amounts to base cations pools which were extractable with 0.01 M $HNO_3$, and conducted repeated sequential $HNO_3$ extractions over periods of hours to days. They observed that base cation pools extracted with 0.01 M $HNO_3$ in 2 hours were similar in magnitude to $BaCl_2$ exchangeable pools, but that in repeated extractions with $HNO_3$, similar amounts of base cations continued to be extractable, while the size of $BaCl_2$ exchangeable pools decreased markedly after the first extraction. Morisuka et al. (2002) observed discreet patterns of bioavailability of

the dilute 0.01 M HCl-extractable potassium pool from the exchangeable pool; during the course of a 17-day growth experiment with Zea mays, both soil pools contributed to plant K, but the HCl-extractable pool exhibited much shorter depletion zones around mays roots than the exchangeable pool, suggesting a more-active role by plants in uptake from the acid-extractable pool. Ortas et al. (1999) grew Italian grass in pots for 6 months and measured the exchangeable and acid-extractable K and Mg contents of the soil before and after; they observed that, on average, ~45% and ~20% of plant K and

Mg contents, respectively, were derived from the acid-extractable, non-exchangeable soil pools. Mengle and Rahmatullah (1994) grew a number of crops in soils of varying exchangeable K availability, and observed that on the soils poorest in exchangeable K, the majority of plant K derived from acid-extractable nonexchangeable pools. Taken together these and other findings clearly indicate that base cation pools that are extractable with moderately concentrated strong acids, and not extractable with salt solutions used to assay the exchangeable base cation pools, may be readily available for plant uptake

and in some equilibrium with exchangeable pools. However, more research is needed to determine what controls the availability of acid-extractable non-exchangeable pools and the rate of flux into exchangeable pools.

Simonsson et al., (2016) used K/Rb ratios in different soil extracts to examine the flux rates between HCl-extractable putatively clay-interlayer K and exchangeable K, and concluded that there was active and continuous exchange between the exchangeable and the "nonexchangeable" HCl-extractable K pools during a single forest rotation. A number

of studies have used stable or radio-isotope tracers in plant-soil systems to examine the relative contributions of different soil pools to plant uptake. Blum et al. (2002) used Ca/Sr and $^{87}Sr/^{86}Sr$ isotope ratios to conclude that apatite minerals were a particularly important source of Ca to ectomycorrhizal trees colonized by ectomycorrhizal fungi. Isotopic dilution experiments provide a promising tool to quantify source and sink reservoirs in the soil and estimate the rates of elemental flux between different soil pools. Newbould and Russell (1963) observed that $^{45}Ca$ added to pots equilibrated rapidly with

the exchangeable Ca in the soil, but when they grew ryegrass in those pots, plants accessed additional Ca from a pool that was not exchangeable. The isotopic dilution of $^{44}Ca$ and $^{26}Mg$ tracers added to a Ca- and Mg-poor beech forest in central France showed that there may be important Ca sources in the soil that are not accounted for by conventional mass-balance approaches (Van der Heijden et al. 2014). Isotopic dilution experiments carried out in the lab on unlabelled soil from the same beech forest showed that the oxalic-acid extractable and the $HNO_3$-extractable pools contributed directly to the labile

pools of Mg, Ca and K. They concluded that these acid-extractable, non-exchangeable pools of Mg, Ca and K are likely to be significant to nutrient cycling on the scale of years to decades. Though isotopic approaches enable the study of elemental fluxes between extractable soil pools, but they are insufficient to better characterize the physicochemical nature of these pools. As discussed above, this challenging question requires the development and combination of new methods such as spectroscopic methods (*e.g.* micro, nanoSIMS, STXM, etc.) to better understand what organic, mineral and

organomineral phases are targeted by the different extraction reactants in order to determine the chemical nature of these pools.





**5. Conclusions**

Our mass-balance estimates indicate that stem-only harvesting would moderately deplete and whole-tree harvesting would markedly deplete exchangeable cation pools, such that current 0.1 M $BaCl_2$ exchangeable pools of base cations are insufficient to support more than a single forest rotation under whole-tree harvesting of the Kindla forest. A large fraction

of these salt extractable base cation pools were found in the organic rich horizons, especially the humus layer. HCl-extractable base cation pools are much larger than the salt exchangeable pools, and if available for plant uptake and/or exchange with the exchangeable pool could sustain many rotations of stem-only or whole-tree harvesting. The large pools of 0.2 M HCl-extractable base cations we observed may be comprised of a combination of microbial biomass, fine-grained easily weatherable minerals that may be protected from weathering by organic matter or secondary mineral coatings, as

well as by the presence of significant stores of non-structural clay interlayer bound K. We were, however, unable to apportion the HCl-extractable base cation pool between these potential pools or draw any conclusions about their flux rates or bioavailability. Specific extraction techniques, including $H_2O_2$ or other oxidative treatments that are specific to organic matter, sodium tetraphenylboron to extract clay-interlayer K without dissolving minerals, as well as more dilute (0.01 M) acid extractions, may help us to more diagnostically estimate the size of different base cation pools. Spectroscopic

techniques may allow us to identify the chemical nature of these pools, while isotopic techniques may allow the quantification of base cation flux rates between exchangeable and acid-extractable pools.

Many soils appear to have very large, relative to exchangeable pools, base cation reserves in the mineral soil which are not extractable with conventional exchangeable cation assays, but which appear to be available for plant uptake and interact with the exchangeable pool on the scale of years to decades. These large pools should be addressed in

ecosystem mass-balance research and accounted for and considered in nutrient management recommendations. To develop forest management policy based on these putatively available pools, a better understanding of their physicochemical nature, bioavailability and flux rates is needed.

**Author contribution**

NPR, KB, and SL conceived the idea for this manuscript. LL, SL, and EM contributed data. NPR analyzed the data and wrote manuscript. All authors contributed ideas, participated in dicussions about the manuscript and with writing and interpretation.

**Competing interests**

The authors are not aware of any competing issues with regards to this article and it's publication.

**Acknowledgements**

The authors would like to thank the funding agencies responsible for this work; The Swedish Research council for funding the research consortion: Quantifying weathering Rates for Sustainable Forestry; The European Commission and the

International Cooperative Programme on Assessment and Monitoring of Air Pollution Effects on Forests (ICP Forests) for providing support for data collection and for the SoilTREC project.



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





Table 1: Sample description for 9 samples from 3 plots, 3 depths at each plot, along the hydrology gradient at Kindla.

| Soil class | Slope position | Hydrology | Soil sampling depth (cm) | Soil horizon | pH (H₂O) | C (% dw) | N (% dw) | Clay (%) (d < 0.002 mm) | Silt (%) (0.002 ≤ d < 0.02 mm) | Sand (%) (0.02 ≤ d < 2 mm) | Site Index (m) |
|---|---|---|---|---|---|---|---|---|---|---|---|
| Podzol | upslope | Groundwater recharge | 0-10 | E-hor | 4.42 | 0.45 | 0.036 | 1.0 | 18 | 81 | 23 |
| Podzol | upslope | Groundwater recharge | 10-20 | B-hor | 4.92 | 2.11 | 0.094 | 2.6 | 20 | 77 | 23 |
| Podzol | upslope | Groundwater recharge | 60-70 | C-hor | 4.91 | 0.43 | 0.028 | 1.0 | 14 | 85 | 23 |
| Regosol | midslope | Groundwater recharge | 5-15 | E-hor | 4.7 | 0.85 | 0.055 | 1.6 | 18 | 81 | 24 |
| Regosol | midslope | Groundwater recharge | 20-30 | B-hor | 4.91 | 2.47 | 0.128 | 2.2 | 18 | 80 | 24 |
| Regosol | midslope | Groundwater recharge | 50-60 | C-hor | 5.15 | 0.91 | 0.043 | 1.6 | 16 | 82 | 24 |
| Regosol | downslope | Groundwater discharge | 15-20 | E-hor | 4.18 | 15.20 | 0.783 | no data | no data | no data | 25 |
| Regosol | downslope | Groundwater discharge | 40-45 | B-hor | 4.72 | 3.36 | 0.167 | 2.1 | 14 | 85 | 25 |
| Regosol | downslope | Groundwater discharge | 55-60 | C-hor | 4.94 | 1.74 | 0.099 | 2.0 | 10 | 89 | 25 |

Table 2: Mass-balances for the three soils collected along the hydrological gradient at Kindla based on annual fluxes of Ca, Mg, K, and Na (mg m⁻² yr⁻¹).

| (mg/m²/yr) | upslope podzol | | | | midslope regosol | | | | downslope regosol | | | |
|---|---|---|---|---|---|---|---|---|---|---|---|---|
| | Ca | K | Mg | Na | Ca | K | Mg | Na | Ca | K | Mg | Na |
| weathering | 291 | 230 | 32 | 623 | 291 | 230 | 32 | 623 | 291 | 230 | 32 | 623 |
| deposition | 163 | 154 | 64 | 493 | 163 | 154 | 64 | 493 | 163 | 154 | 64 | 493 |
| leaching | 114 | 152 | 100 | 945 | 159 | 82 | 97 | 976 | 259 | 82 | 133 | 976 |
| uptake, stem only | 219 | 121 | 30 | 6.5 | 219 | 121 | 30 | 6.5 | 219 | 121 | 30 | 6.5 |
| uptake, whole tree | 538 | 311 | 69 | 16 | 538 | 311 | 69 | 16 | 538 | 311 | 69 | 16 |
| balance, stem only | 122 | 111 | **-34** | 165 | 77 | 181 | **-31** | 134 | **-23** | 181 | **-67** | 134 |
| balance, whole tree | **-198** | **-79** | **-73** | 155 | **-243** | **-9** | **-70** | 124 | **-343** | **-9** | **-106** | 124 |

Mass balances in bold are negative, indicating a net loss of base cations from the system.





Table 3: Number of forest harvest rotations (65 yr) each base cation pool may offset net ecosystem losses for for each of three soils. All extractions were performed on the mineral soil; for BaCl$_2$ and Aqua Regia extractions, humus samples from nearby soils were seperately sampled and anlyzed, and these humus-derived pools have been added to the mineral soil pools for comparison.

| | upslope podzol | | | | midslope regosol | | | | downslope regosol | | | |
|---|---|---|---|---|---|---|---|---|---|---|---|---|
| **H$_2$O extractable mineral soil** | | | | | | | | | | | | |
| | Ca | K | Mg | Na | Ca | K | Mg | Na | Ca | K | Mg | Na |
| mg/m$^2$ | 610 | 2536 | 390 | 1307 | 1179 | 1104 | 558 | 1668 | 2564 | 3803 | 1532 | 5964 |
| rot SO | + | + | 0.2 | + | + | + | 0.3 | + | 1.7 | + | 0.4 | + |
| rot WT | 0.0 | 0.5 | 0.1 | + | 0.1 | 1.8 | 0.1 | + | 0.1 | 6.2 | 0.2 | + |
| **BaCl$_2$ extractable mineral soil** | | | | | | | | | | | | |
| | Ca | K | Mg | Na | Ca | K | Mg | Na | Ca | K | Mg | Na |
| mg/m$^2$ | 3135 | 2938 | 709 | 958 | 11953 | 1244 | 1395 | 912 | 32474 | 3234 | 5832 | 4129 |
| rot SO | + | + | 0.3 | + | + | + | 0.7 | + | 21.3 | + | 1.3 | + |
| rot WT | 0.2 | 0.6 | 0.1 | + | 0.8 | 2.0 | 0.3 | + | 1.5 | 5.3 | 0.8 | + |
| **BaCl$_2$ extractable mineral soil + humus** | | | | | | | | | | | | |
| | Ca | K | Mg | Na | Ca | K | Mg | Na | Ca | K | Mg | Na |
| mg/m$^2$ | 17657 | 11636 | 6745 | 1819 | 22942 | 9493 | 4971 | 1416 | 43553 | 19277 | 11178 | 5539 |
| rot SO | + | + | 3.1 | + | + | + | 2.5 | + | 28.6 | + | 2.6 | + |
| rot WT | 1.4 | 2.3 | 1.4 | + | 1.5 | 15.5 | 1.1 | + | 2.0 | 31.4 | 1.6 | + |
| **0.2 M HCl extractable mineral soil** | | | | | | | | | | | | |
| | Ca | K | Mg | Na | Ca | K | Mg | Na | Ca | K | Mg | Na |
| mg/m$^2$ | 106221 | 42055 | 176304 | 29931 | 104022 | 40035 | 219370 | 37761 | 156633 | 52285 | 264582 | 31826 |
| rot SO | + | + | 80 | + | + | + | 110 | + | 103 | + | 61 | + |
| rot WT | 8 | 8 | 37 | + | 7 | 65 | 48 | + | 7 | 85 | 38 | + |
| **0.5 M HCl extractable mineral soil** | | | | | | | | | | | | |
| | Ca | K | Mg | Na | Ca | K | Mg | Na | Ca | K | Mg | Na |
| mg/m$^2$ | 204944 | 56492 | 256499 | 39574 | 175093 | 53303 | 295772 | 50811 | 244148 | 63845 | 307574 | 42683 |
| rot SO | + | + | 117 | + | + | + | 148 | + | 160 | + | 71 | + |
| rot WT | 16 | 11 | 54 | + | 11 | 87 | 65 | + | 11 | 104 | 45 | + |
| **Aqua Regia extractable mineral soil** | | | | | | | | | | | | |
| | Ca | K | Mg | Na | Ca | K | Mg | Na | Ca | K | Mg | Na |
| mg/m$^2$ | 1266310 | 124126 | 478371 | 44963 | 1022389 | 111295 | 504097 | 44278 | 1276096 | 130620 | 496495 | 40348 |
| rot SO | + | + | 218 | + | + | + | 252 | + | 838 | + | 115 | + |
| rot WT | 98 | 24 | 100 | + | 65 | 182 | 110 | + | 57 | 213 | 72 | + |
| **Aqua Regia extractable mineral soil + humus** | | | | | | | | | | | | |
| | Ca | K | Mg | Na | Ca | K | Mg | Na | Ca | K | Mg | Na |
| mg/m$^2$ | 1296274 | 139710 | 489987 | 46228 | 1046073 | 127455 | 512476 | 45400 | 1306294 | 167864 | 512194 | 44366 |
| rot SO | + | + | 221 | + | + | + | 255 | + | 847 | + | 116 | + |
| rot WT | 100 | 26 | 102 | + | 65 | 195 | 111 | + | 58 | 243 | 73 | + |

"+" Indicates no net loss;
"rot SO"  is the number of 65-year rotations of stem-only harvest
"rot WT"  is the number of 65-year rotations of whole-tree harvest

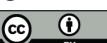



Table 4 Mineral soil base cation concentrations from different soil fractions as defined by extractant and relative molar fraction of each base cation within each extraction

| 0.1 M BaCl₂ (mg/kg dw) | | | | | Molar Fractional abundance | | | BCₓ/ BC_tot |
|---|---|---|---|---|---|---|---|---|
| Soil sample | Ca | K | Mg | Na | Ca | K | Mg | Na |
| upslope E_horizon | 7.4 | 8.3 | 2.3 | 2.5 | 31% | 35% | 16% | 18% |
| upslope B_horizon | 5.7 | 10.9 | 2.7 | 3.8 | 20% | 40% | 16% | 24% |
| upslope C_horizon | 1.7 | 4.0 | 0.6 | 1.3 | 18% | 46% | 10% | 25% |
| midslope E_horizon | 13.4 | 3.4 | 4.3 | 1.9 | 49% | 13% | 26% | 12% |
| midslope B_horizon | 28.8 | 3.4 | 4.9 | 3.6 | 62% | 8% | 17% | 13% |
| midslope C_horizon | 14.9 | 3.0 | 1.9 | 1.7 | 62% | 13% | 13% | 12% |
| downslope E_horizon | 92.1 | 17.4 | 32.7 | 24.4 | 45% | 9% | 26% | 21% |
| downslope B_horizon | 30.8 | 3.8 | 7.3 | 4.2 | 57% | 7% | 22% | 13% |
| downslope C_horizon | 27.5 | 4.5 | 6.2 | 4.2 | 55% | 9% | 21% | 15% |

| Potential Microbial Biomass (mg/kg dw) | | | | | Molar Fractional abundance | | | BCₓ/ BC_tot |
|---|---|---|---|---|---|---|---|---|
| | Ca | K | Mg | Na | Ca | K | Mg | Na |
| upslope E_horizon | 0.44 | 2.15 | 0.07 | 0.05 | 15% | 78% | 4% | 3% |
| upslope B_horizon | 2.03 | 10.00 | 0.34 | 0.21 | 15% | 78% | 4% | 3% |
| upslope C_horizon | 0.41 | 2.02 | 0.07 | 0.04 | 15% | 78% | 4% | 3% |
| midslope E_horizon | 0.81 | 4.00 | 0.14 | 0.08 | 15% | 78% | 4% | 3% |
| midslope B_horizon | 2.38 | 11.71 | 0.40 | 0.25 | 15% | 78% | 4% | 3% |
| midslope C_horizon | 0.87 | 4.29 | 0.15 | 0.09 | 15% | 78% | 4% | 3% |
| downslope E_horizon | 14.62 | 72.04 | 2.45 | 1.52 | 15% | 78% | 4% | 3% |
| downslope B_horizon | 3.23 | 15.92 | 0.54 | 0.34 | 15% | 78% | 4% | 3% |
| downslope C_horizon | 1.67 | 8.25 | 0.28 | 0.17 | 15% | 78% | 4% | 3% |

| 0.2 M HCl only (0.2 M HCl - BaCl₂- mic. bio.) (mg/kg dw) | | | | | Molar Fractional abundance | | | BCₓ/ BC_tot |
|---|---|---|---|---|---|---|---|---|
| | Ca | K | Mg | Na | Ca | K | Mg | Na |
| upslope E_horizon | 32 | 108 | 13 | 135 | 8% | 28% | 5% | 59% |
| upslope B_horizon | 88 | 52 | 563 | 45 | 8% | 5% | 81% | 7% |
| upslope C_horizon | 212 | 136 | 814 | 49 | 12% | 8% | 75% | 5% |
| midslope E_horizon | 56 | 92 | 173 | 107 | 9% | 15% | 46% | 30% |
| midslope B_horizon | 152 | 78 | 773 | 103 | 9% | 5% | 76% | 11% |
| midslope C_horizon | 210 | 118 | 780 | 81 | 12% | 7% | 73% | 8% |
| downslope E_horizon | 118 | 10 | 227 | 59 | 20% | 2% | 62% | 17% |
| downslope B_horizon | 125 | 86 | 566 | 89 | 10% | 7% | 72% | 12% |
| downslope C_horizon | 162 | 196 | 1253 | 70 | 6% | 8% | 81% | 5% |

| 0.5 M HCl only (0.5 M HCl - 0.2 M HCl) (mg/kg dw) | | | | | Molar Fractional abundance | | | BCₓ/ BC_tot |
|---|---|---|---|---|---|---|---|---|
| | Ca | K | Mg | Na | Ca | K | Mg | Na |
| upslope E_horizon | 26 | 29 | 9 | 38 | 19% | 21% | 11% | 48% |
| upslope B_horizon | 105 | 27 | 202 | 17 | 21% | 6% | 67% | 6% |
| upslope C_horizon | 198 | 58 | 424 | 21 | 20% | 6% | 70% | 4% |
| midslope E_horizon | 54 | 39 | 52 | 53 | 20% | 15% | 31% | 34% |
| midslope B_horizon | 121 | 25 | 213 | 14 | 23% | 5% | 67% | 5% |
| midslope C_horizon | 156 | 42 | 340 | 37 | 19% | 5% | 68% | 8% |
| downslope E_horizon | 46 | 25 | 106 | 30 | 15% | 9% | 59% | 17% |
| downslope B_horizon | 84 | 30 | 47 | 38 | 32% | 12% | 30% | 26% |
| downslope C_horizon | 134 | 36 | 187 | 18 | 26% | 7% | 60% | 6% |

| Aqua Regia only (Aq. Reg. - 0.5 M HCl) (mg/kg dw) | | | | | Molar Fractional abundance | | | BCx/BC_tot |
|---|---|---|---|---|---|---|---|---|
| | Ca | K | Mg | Na | Ca | K | Mg | Na |
| upslope E_horizon | 287 | 65 | 114 | -77 | 70% | 16% | 46% | -33% |
| upslope B_horizon | 1381 | 232 | 719 | 50 | 48% | 8% | 41% | 3% |
| upslope C_horizon | 1998 | 238 | 923 | 70 | 51% | 6% | 39% | 3% |
| midslope E_horizon | 517 | 108 | 212 | -74 | 61% | 13% | 41% | -15% |
| midslope B_horizon | 1387 | 137 | 542 | 11 | 57% | 6% | 37% | 1% |
| midslope C_horizon | 1978 | 215 | 896 | 12 | 54% | 6% | 40% | 1% |
| downslope E_horizon | 651 | 140 | 279 | -5 | 52% | 12% | 37% | -1% |
| downslope B_horizon | 893 | 151 | 396 | -43 | 55% | 10% | 40% | -5% |
| downslope C_horizon | 1621 | 238 | 822 | 29 | 50% | 7% | 41% | 2% |



Figure 1: Extractable contents of calcium (a), potassium (b), magnesium (c), and sodium (d) across 3 soils and 3 soil horizons obtained form six different chemical extractions. Note the $\log_{10}$ scale on Y-axis.

Figure 2: Base cation pool sizes of calcium (a), potassium (b), magnesium (c), and sodium (d) across 3 soils and 3 soil horizons obtained from soil extraction ($BaCl_2$ extractions), differences between extractions (Aqua Regia, 0.5 M HCl, and 0.2 M HCl extractants) and extrapolated estimates (Mic_Bio 10 – microbial biomass; estimated from SOC contents).







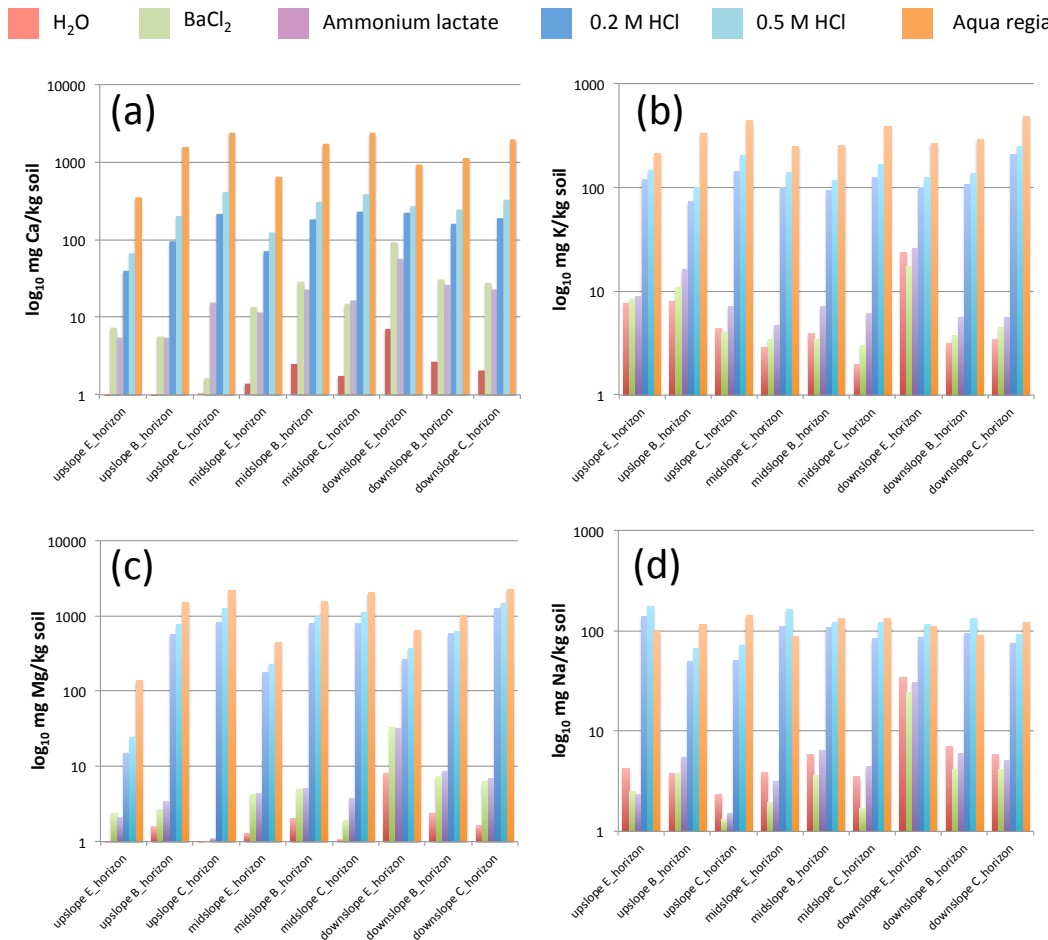

Figure 1



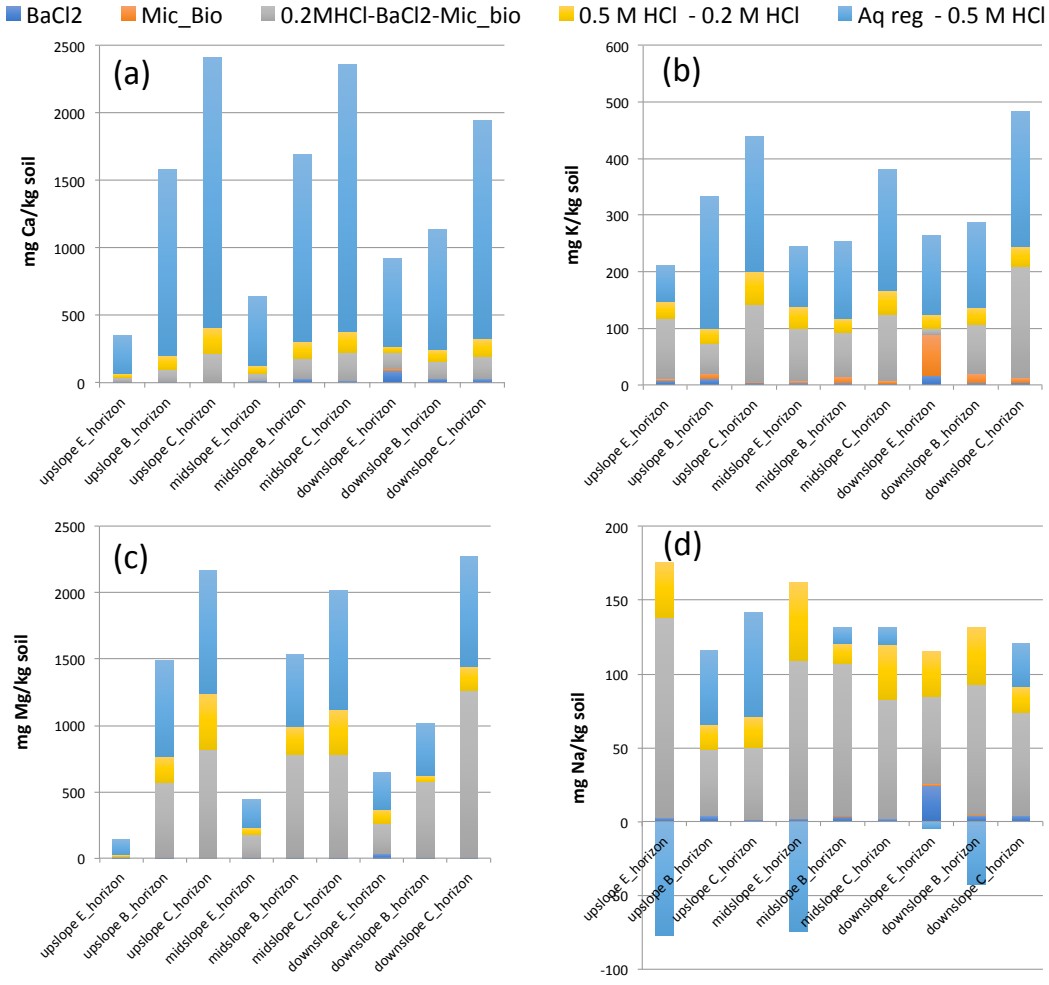

Figure 2