# Peer review of "Base cations in the soil bank. Non-exchangeable pools may sustain centuries of net loss to forestry and leaching."

_SOIL, 2019_

## Referee Comment (RC1) · Anonymous Referee #1 · 21 May 2019

MS No.: Soil-2019-5 SOIL

Title: Base cations in the soil bank. Non-exchangeable pools may sustain centuries of net loss to forestry and leaching Author(s): Nicholas P. Rosenstock et al. MS Type: Short communication Special Issue: Quantifying weathering rates for sustainable forestry (BG/SOIL inter-journal SI)

General Comments

The paper examines different methods for quantifying extractable base cations in a forest soil. The different methods give different pool sizes of extractable elements, which is not surprising, but the careful comparison to other data from the forest makes

for a nice read. Comparative studies are always helpful to readers.

My biggest question is whether you did a so-called total element extraction, which I believe is some combination of hydrofluoric acid (HF), perchloric acid (HClO4), and nitric acid (HNO3)? It seems necessary to define the upper bound of elements, the total elemental pool, to put the extractable pool size into perspective.

Overall, the paper is well written.

Specific Comments

1) The Abstract reads well.

2) The Introduction section reads well. My only comment is what about Mehlich-3 extraction. This is common in may soil fertility studies. How does it compare? I am not suggesting that you provide a complete literature review in the Introduction. However, keep in mind that the literature is filled with ways to assess extractable nutrients.

1) The Methods are okay, to me. I do have a few questions, however. Does the length of time for the HCl-extraction matter? It seems that the extractable amount would depend upon time for different pools to equilibrate with the acid. Also, did the Aqua Regia include several additions of hydrogen peroxide? Was the Aqua Regia done with hotplate or microwave? As a I understand, there are several variations of Aqua Regia, which tend to give different extractable amounts.

3) Using a 65-year rotation is fine. However, this value seemed to be lost in text. Indeed, I did not realize that you had specified age until the Results. Perhaps, the paragraph including page 5, line 6 could be reorganized a bit to emphasize the assumption.

4) Table 2 is a bit odd, to me, because the differences among sites depends only on leaching. I find it surprising the same values apply for weathering among the three sites. Is this real, or a best guess?

5) I suppose that I should read the supplemental methods, but your calculation of microbial biomass pool is confusing to me. Why not do the typical fumigation-extraction? I believe Anderson and Domsch (Anderson, J. P. E., & Domsch, K. H. (1980). Quantities of plant nutrients in the microbial biomass of selected soils. Soil Science, 130(4), 211-216) did so.

6) The same sort of logic applies to base cations bound to organic matter. Why not do the typical before and after hydrogen peroxide treatment? Or, did you try a chelating agent, such as citric acid and ethylenediaminetetraacetic acid (EDTA)?

7) While I agree that the calcium in the calcium oxalate pool is not large, are you sure that you got all of the calcium bound to humic compounds?

8) With all do apologizes, I think that the section on Primary Minerals (3.4.6) is a bit confusing. The literature is filled with studies that assess base cations in primary minerals and, as mentioned above, most use some combination of HF, HClO4, HNO3. Without this measure, I simply do not understand how you can claim 0.2 M HCl is good enough. I seem to be missing the point here.

9) The Discussion section is okay.

Technical Comments

2) Page 1, Line 15: change 'pools' to 'pool size'.

3) Page 2, line 5: consider saying, 'because biomass extraction also removes nutrient elements in biomass.'

4) Page 2, line 37: 'barium and ammonium salts' or 'barium salts or ammonium salts'?

5) Page 3, line 1: perhaps delete 'insoluble': 'complexes' does not need a modifier.

6) Page 8, line 7: change 'e.g.' to 'for example.'

7) Page 8, line 8: cations also form complexes with humic compounds.

8) Page 8, line 16: does 0.2 M HCl dissolve organic compounds?

9) References: check the list carefully. There are duplicates in a few places.

---

## Referee Comment (RC2) · Anonymous Referee #2 · 20 Jul 2019

In relation to biomass harvest, especially whole tree harvesting, I'm also interested in quantification of sustainable levels of biomass harvesting or soil productivity without nutrient inputs in forest ecosystem. By the subtraction of each extraction of base cations, it is meaningful to demonstrate the sources of base cations or availability for plant uptake. In addition, as the report of the potential size of soil nutrient pools in base cations is fewer than that in nitrogen, the data of soil base cation concentration from different soil fraction in each horizon are valuable.

However, soil sampling depth was discrete and not continuous from surface soil to lower in each soil pit. Moreover, soil bulk density was not shown in each depth. Therefore, estimation of all amounts of soil base cation was unsatisfied with the prerequisites of mass-balance method and improper as utilization of mass-balance method; this estimation was not able to indicate correctly the all amounts of base cations in each soil pit although comparison of the amounts and trends of base cations among the horizons in each soil would be possible from surface soil to lower or from upslope soil to downslope. Furthermore, if the comparison of uptake fluxes of base cations among the soil pits with the difference of the hydrological gradient, as in the difference of moisture condition of the soil pits, amounts of uptake of base cations by plants should be different among the soil pits, you should examine the uptake by the soil pit.

Finally, although this manuscript provides some interesting scientific results, due to misuse of mass balance method and rough estimation mentioned above, it is difficult to accept for publication in Soil in this time, and I suggest to submit to other journals or resubmit to.

---

## Author Comment (AC1) · 19 Aug 2019

Responses to reviewer MS No.: Soil-2019-5 SOIL

Anonymous Referee #1 General Comments The paper examines different methods for quantifying extractable base cations in a forest soil. The different methods give different pool sizes of extractable elements, which is not surprising, but the careful comparison to other data from the forest makes for a nice read. Comparative studies are always helpful to readers. My biggest question is whether you did a so-called total element extraction, which I believe is some combination of hydrofluoric acid (HF), perchloric acid (HClO4), and nitric acid (HNO3)? It seems necessary to define the upper bound

of elements, the total elemental pool, to put the extractable pool size into perspective.

No complete digestion was performed on these soils; the most aggressive digestion was with Aqua Regia being a common pseudototal digestion procedure (ISO11466:1995) with a legal status in some European countries. It has also been used as a reference procedure in the preparation of soil reference materials certified for extractable contents by the European Community of Bureau of Reference. The nutrients that are not digested with Aqua Regia belong primarily to the more recalcitrant fractions of silicate minerals (Rao et al., 2008; Nezat et al., 2007), and likely would not have weathered appreciably over the next millennia.

The true upper bound of nutrient contents, determined after HF or LiBo-fusion digestion, would have added valuable perspective and if we had had the resources, we would have liked to add a complete digestion to the set of extractions. However, our focus was on different soil nutrient fractions, the availability of which was relevant to forest growth in the centuries to millenia time-scale. The minerals which do completely dissolve in Aqua Regia, like apatite and biotite, are comparatively unabundant compared to the feldspars which dominate the mineralogy at these sites, but make an outsize contribution to nutrient reserves in the short to medium term. Hence, we do not think the absence of total element extraction data detracts from our comparison of different nutrient pool sizes to nutrient depletion over time in different forest management scenarios.

Overall, the paper is well written. Specific Comments 1) The Abstract reads well. 2) The Introduction section reads well. My only comment is what about Mehlich-3 extraction. This is common in may soil fertility studies. How does it compare? I am not suggesting that you provide a complete literature review in the Introduction. However, keep in mind that the literature is filled with ways to assess extractable nutrients.

Except for Aqua Regia we avoided composite extraction solutions, particularly ones that were designed, as Mehlich -3 was, to extract from a broad suite of potential binding sites in the soil.

1) The Methods are okay, to me. I do have a few questions, however. Does the length of time for the HCl-extraction matter? It seems that the extractable amount would depend upon time for different pools to equilibrate with the acid.

We agree that the extraction time is an important factor, and the extraction yields from the HCl digestions were likely to have been especially sensitive to incubation time. The incubation period was standardized for each method, and all extractions, except for Aqua Regia, were performed for 1.5 – 2 hours; Aqua Regia digestion was performed over a 14-hour time period. The time periods for all extractions are noted in table S1 in the supplementary methods section.

2) Also, did the Aqua Regia include several additions of hydrogen peroxide? Was the Aqua Regia done with hotplate or microwave? As I understand, there are several variations of Aqua Regia, which tend to give different extractable amounts.

Our Aqua Regia digestion was performed on a hot plate and did not involve predigestion, with either hydrogen peroxide or additional HNO3, as our protocol recommends for samples of higher organic matter content ( > 20%). We noted in the supplementary methods that the Aqua Regia digestion was performed according to ISO 11466:1995, and have now, in light of the reviewer's query added some information to clarify the basic procedures of this method.

3) Using a 65-year rotation is fine. However, this value seemed to be lost in text. Indeed, I did not realize that you had specified age until the Results. Perhaps, the paragraph including page 5, line 6 could be reorganized a bit to emphasize the assumption.

We thank the reviewer for the recommendation. We had also noted in section 2.3 of the methods the use of a 65 year time period to represent the forest rotation period. However, given the importance of this metric to our presentation of the results we agree that a more clear presentation would be advisable, and as such we have now added the sentence in the beginning of section 3.3: "For each harvest scenario–element-soil

combination under which a net ecosystem loss was predicted we calculated the number of 65-year harvest rotations that each extractant-defined base cation pool could offset."

4) Table 2 is a bit odd, to me, because the differences among sites depends only on leaching. I find it surprising the same values apply for weathering among the three sites. Is this real, or a best guess?

The reviewer is correct in his/her observation. We endeavored to make this clear in section 2.2 of the methods, where we noted that: "Deposition, weathering, and biomass uptake fluxes were available for the entire catchment including the three soil sampling sites located along a hydrological gradient (recharge, intermediate and discharge areas), while leaching losses were calculated for each of these three hydrological locations" We have now changed the word "available" to "calculated" to make this more clear.

Weathering was modeled using the PROFILE model, which is, as any weathering rate estimate is, a "best guess", albeit one that has been developed over decades of research. The model was run based on soil data primarily from the upslope and midslope locations of our gradient. We understand that having so many values repeated 3 times may look "odd" as the reviewer notes, but we still think that the table, which shows how the different elements, differences in leaching, and the differences in harvest removals result in a range of mass-balance results is valuable in its present form. In addition, as we have soil extraction data for each site, the "results" of table 2 are the inputs for the calculations we present in table 3, and it thus seems valuable to explicitly show what they are based on.

5) I suppose that I should read the supplemental methods, but your calculation of microbial biomass pool is confusing to me. Why not do the typical fumigation-extraction? I believe Anderson and Domsch (Anderson, J. P. E., & Domsch, K. H. (1980). Quantities of plant nutrients in the microbial biomass of selected soils. Soil Science, 130(4),

211-216) did so.

We did not have the resources to or the intention of measuring microbial biomass contents when we collected the soils. In addition, the challenges of devising standard curves for extraction efficiency of the different elements across the different soils for elements that are not typically examined in chloroform-fumigation assays (where C, N and P are the most commonly examined elements) were considerable. In this work, the goal of the estimation of the potential microbial biomass contents was to examine whether microbial biomass could contribute significantly to the extraction yields obtained from the more aggressive acid extractions, and, as such, an accuracy range of +/- 50%, which we are well under, is quite acceptable. The basic premise of this estimation, that microbial biomass carbon is generally about 1% of soil carbon, has shown itself to be a quite robust, if somewhat rough, estimate. In short, we did not have the resources for chloroform fumigation and we think the estimation method we used is appropriate for the level of precision we needed.

6) The same sort of logic applies to base cations bound to organic matter. Why not do the typical before and after hydrogen peroxide treatment? Or, did you try a chelating agent, such as citric acid and ethylenediaminetetraacetic acid (EDTA)?

We did not perform a hydrogen peroxide pre-treatment or employ any chelating agents. The Aqua Regia extraction completely dissolved the organic matter, and as we note in section 3.1 of the results (p.5, ln. 27) "Humus layer 0.1 M BaCl2 extractable pools of Ca, K, Mg, and Na were 88, 100, 86, and 99%, respectively, as large as Aqua Regia extracted pools, indicating that nearly all of the humus-associated base cation pools are exchangeable." We discuss this also in the discussion in section 3.4.3 ("Base cations bound to soil organic matter"). We think that organic matter may, in particular, have a role in shielding mineral surfaces from acid attack, and conclude this section by noting that in future research "chemical treatments utilizing hydrogen peroxide or surfactants (Chao, 1984) could be used to remove or reduce such coatings without the use of a strong acid"

7) While I agree that the calcium in the calcium oxalate pool is not large, are you sure that you got all of the calcium bound to humic compounds?

No, we cannot be sure that we got all of the calcium bound to humic compounds in the less aggressive extractions, but we are fairly certain that we did in the Aqua Regia extraction, and as we noted above nearly all of the monovalent, and over 86% of the bivalent cations in the humus layer were solubilized by the BaCl2 extraction. We assumed that Ca-oxalate would be resistant to BaCl2 extractions but would solubilize in the 0.2 and 0.5 M HCl extractions, based in part on the protocols developed by Dauer and Perakis (2013; 2014) . This is why we examine the discrepancy between different Ca fractions as a function of depth and organic matter. Observing that the 0.2 M HCl –only fractions of Ca do not correlate with organic carbon content, and increase with depth, rather than decrease as would be expected if Ca-oxalate were a major pool of HCl-only solubilized Ca, are, together, fairly compelling evidence that Ca-oxalate does not make a large contribution to the acid-extractable Ca pools. We cannot, with our methodology, determine how small a contribution it makes.

8) With all do apologizes, I think that the section on Primary Minerals (3.4.6) is a bit confusing. The literature is filled with studies that assess base cations in primary minerals and, as mentioned above, most use some combination of HF, HClO4, HNO3. Without this measure, I simply do not understand how you can claim 0.2 M HCl is good enough. I seem to be missing the point here.

Section 3.4 examines the potential sources (physicochemical origin) of base cations in the very large HCl-extractable pools. Section 3.4.6 examines the potential for primary mineral dissolution to have made a significant contribution to the large HCl-extractable pools. These rather weak acid-extractions were intended to access non-exchangeable pools that may make a significant contribution to nutrient cycling on a decadal time scale. We were not interested in the total mineral contents because of the much longer time scale their total contents are relevant for. We did however also perform Aqua Regia digestion which yielded on average an order of magnitude greater elemental yields

than the 0.5 M HCl extraction and these values are useful for comparison to the HCl extractions, both as a highly conservative estimate of pseudototal elemental contents (as silicate rocks are only incompletely dissolved) and as an indication of what the primary mineral elemental rations are (as we can assume that the Aqua Regia only elemental rations will more closely approximate the mineral elemental rations than the HCl-only elemental ratios). We have modified section 3.4 so that the first sentence reads "Given the large size and potential to buffer against leaching and many forest harvest rotations of net base cation loss, understanding the chemical nature and availability of the HCl-extractable pool is important to forming management recommendations. "

9) The Discussion section is okay.

Technical Comments 2) Page 1, Line 15: change 'pools' to 'pool size'. Changed

3) Page 2, line 5: consider saying, 'because biomass extraction also removes nutrient elements in biomass.' Changed

4) Page 2, line 37: 'barium and ammonium salts' or 'barium salts or ammonium salts'? Changed

5) Page 3, line 1: perhaps delete 'insoluble': 'complexes' does not need a modifier. We agree with the reviewer and have changed the sentence to: "Calcium is known to form strong complexes with organic compounds such as the oxalate ion, which are poorly soluble and not salt-extractable (Dauer et al. 2014)."

6) Page 8, line 7: change 'e.g.' to 'for example.' Changed

7) Page 8, line 8: cations also form complexes with humic compounds.

Yes, we agree, but in this context we are trying to distinguish between physicochemical speciation that is classically viewed as exchangeable and thus extractable with BaCl2, and other organic complexes which are not e.g. Ca-oxalate. On page 8, line 4 we show that almost all monovalent cations (>99%) are salt extractable and most (86-88%) of the divalent cations are as well. Thus, all monovalent and most divalent ions are much

more weakly bound to the organic matrix (presumably humic compounds) compared with Ca-oxalate complexes.

8) Page 8, line 16: does 0.2 M HCl dissolve organic compounds?

HCl likely does not dissolve many organic compounds, but it is likely to effectively solubilize a large portion of the organically bound Ca. This is why we refer to dissolving the "organically-bound Ca". However we agree that the word "completely" belies the evidence, and, as such, we have changed "entirely" to "largely".

9) References: check the list carefully. There are duplicates in a few places. Thank you for bringing this to our attention. We have reviewed and proofread the references.

References: Dauer, J. M., Perakis, S. S.: Contribution of Calcium Oxalate to Soil-Exchangeable Calcium, Soil Science, 178, 671-678, doi:10.1097/SS.0000000000000029, 2013.

Dauer, J. M., Perakis, S. S.: Calcium oxalate contribution to calcium cycling in forests of contrasting nutrient status, Forest Ecology and Management, 334, 64–73, 2014.

Nezat, C.A., Blum, J.D., Yanai, R.D., Hamburg, S.P.: A sequential extraction to determine the distribution of apatite in granitoid soil mineral pools with application to weathering at the Hubbard Brook Experimental Forest, NH, USA. Appl. Geochem, 22, 11, 2406−2421, 2007.

Rao, C.R.M., Sahuquillo, A., Lopez Sanchez, J.F.: A Review of the Different Methods Applied in Environmental Geochemistry For Single and Sequential Extraction of Trace Elements in Soils and Related Materials. Water Air Soil Pollut 189, 291-333. Doi: 10.1007/s11270-007-9564-0, 2008.

---

## Author Comment (AC2) · 19 Aug 2019

Responses to reviewer MS No.: Soil-2019-5 SOIL

Anonymous Referee #2

In relation to biomass harvest, especially whole tree harvesting, I'm also interested in quantification of sustainable levels of biomass harvesting or soil productivity without nutrient inputs in forest ecosystem. By the subtraction of each extraction of base cations, it is meaningful to demonstrate the sources of base cations or availability for plant uptake. In addition, as the report of the potential size of soil nutrient pools in base

cations is fewer than that in nitrogen, the data of soil base cation concentration from different soil fraction in each horizon are valuable.

However, soil sampling depth was discrete and not continuous from surface soil to lower in each soil pit.

Soil sampling was indeed discrete, but effort was taken to represent each discernable horizon in the soil, and determine an inventory for the entire soil profile. Furthermore sampling depth within each horizon was broad (5-10 cm) to integrate as much as possible. The depth and bulk density of each horizon was taken into account when converting from extraction contents to available pools per unit forest area. We have added to the supplementary methods section to make this more clear. "To calculate the total available nutrient pools on an areal basis, nutrient yields from each extract (mg/kg soil) and each horizon were multiplied by the dry soil mass of that horizon, which is the product of the bulk density and the depth of that horizon, which was measured at the time of sampling."

Moreover, soil bulk density was not shown in each depth.

We thank the reviewer for pointing this out. Bulk density was estimated for each horizon with the pedo-transfer function from Nilsson and Lundin (2006) based on depth and organic carbon content. We have now added bulk density to table 1, and detailed the method used in the supplemental methods section. Actual measurements of bulk density would be preferable, but they were not available at the time of sampling. Given our use of the data, we deem the bulk density values obtained from these pedo-transfer functions to be suitable. This pedo-transfer function was derived from hundreds of Swedish soils, with similar soil types, and the error can be expected to be under 15%. We have now added discussion about the potential variation between the calculated bulk density values and the actual bulk density in the supplemental methods.

Therefore, estimation of all amounts of soil base cation was unsatisfied with the prerequisites of mass-balance method and improper as utilization of mass-balance method;

this estimation was not able to indicate correctly the all amounts of base cations in each soil pit although comparison of the amounts and trends of base cations among the horizons in each soil would be possible from surface soil to lower or from upslope soil to downslope.

While more comprehensive sampling would have improved the precision of our data, we do not think the use of discrete sampling depths and a pedo-transfer function to estimate bulk density stands in the way of our interpolating the discrete observations to estimate the base cation content across the entire soil profile. Therefore we believe that our mass-balance analysis of the soil nutrient pools is valid, with all necessary calculations and assumptions specified. The mass-balance estimates are based on estimated flows into and out of the system, and, as such, are not affected by bulk density or soil depth. (The weathering model also uses bulk density measurements). We examined different extractant-defined nutrient pools and compared their magnitude; these different pools were equally affected by the use of a pedotransfer funtion to estimate bulk density and the use of discretized depths for sampling. The comparison of mass-balances to soil nutrient pools over the course of a year or an entire forest rotation may have been affected by the use of a pedotransfer function to estimate bulk density and the use of discretized depths for sampling, but here our focus was more on how the estimated time periods that different nutrient pools could sustain net ecosystem losses compared to one another, and less on the absolute numbers

Furthermore, if the comparison of uptake fluxes of base cations among the soil pits with the difference of the hydrological gradient, as in the difference of moisture condition of the soil pits, amounts of uptake of base cations by plants should be different among the soil pits, you should examine the uptake by the soil pit.

It was not possible to measure nutrient uptake at each position along the hydrological gradient. What we do have data for is relative growth rate (table 1; site index), and while growth is similar along the gradient, the trees at the lower end of the gradient (downslope groundwater discharge) exhibit the highest growth rates, and those at the highest

end (upslope groundwater recharge) exhibit the lowest growth rates, while those at midslope (groundwater recharge) exhibit intermediate growth rates. The biomass uptake values we have are for the trees at the midslope (site index = 24; table 1). We can thus infer that uptake rates at the downslope location (site index = 25) will be somewhat higher, and those at the upslope position (site index = 23) somewhat lower. Zetterberg et al. (2014) have shown that at this site (Kindla) an increase of site index from 24m to 28m corresponds to an increase in the total biomass from 182 to 193 tonnes ha-1. Hence, increasing the site index by 1m can be expected to affect the biomass by less than 2%. We will add discussion about the potential for this variation in uptake rates to affect our projections of the time period that different nutrient pools can buffer net losses for.

Finally, although this manuscript provides some interesting scientific results, due to misuse of mass balance method and rough estimation mentioned above, it is difficult to accept for publication in Soil in this time, and I suggest to submit to other journals or resubmit to.

We hope that the revisions have presented our methodology more clearly so as to show that we did not "misuse" the mass-balance method. Thus we have hopefully now succeeded in demonstrating that our calculation of pools are not flawed by being based on sampling at discrete depths.

References: Nilsson, T., Lundin, L.: Predictions of bulk density of Swedish forest soils from the organic content and soil depth. Report in Forest Ecology and Forest Soils no 91, Swedish University of Agricultural Sciences, Uppsala. ISSN 0348-3398, 41 pp. (In Swedish, English summary), 2006.

Zetterberg, T., Köhler, S.J., Löfgren, S.: Sensitivity analyses of MAGIC modelled predictions of future impacts of whole-tree harvest on soil calcium supply and stream acid neutralizing capacity. Science of The Total Environment 494–495, 187-201. http://dx.doi.org/10.1016/j.scitotenv.2014.06.114, 2014.